# Identification of Corn Chaff as an Optimal Substrate for the Production of Rhamnolipids in *Pseudomonas aeruginosa* Fermentations

**Adriana Bava [1], Sara Carnelli [1], Mentore Vaccari [2]**, **Trello Beffa [3] and Fabrizio Beltrametti [1,***]

1    BioC-CheM Solutions S.r.l., 21040 Gerenzano, VA, Italy; abava@bioc-chemsolutions.com (A.B.); scarnelli@bioc-chemsolutions.com (S.C.)
2    Dipartimento di Ingegneria Civile, Architettura, Territorio, Ambiente e di Matematica (DICATAM), University Degli Studi of Brescia, 25121 Brescia, BS, Italy; mentore.vaccari@unibs.it
3    MADEP S.A., Via Sacerdote Caroni 2, CH-6862 Rancate, Switzerland; trello.beffa@madep-sa.com
*    Correspondence: fbeltrametti@bioc-chemsolutions.com; Tel.: +39-02-9647-4404

**Abstract:** Waste biomass deriving from agricultural activities has different destinations depending on the possibility of applying it to specific processes. As the waste biomass is abundant, cheap, and generally safe, it can be used for several applications, biogas production being the most relevant from the quantitative point of view. In this study, we have used a set of agricultural by-products (agro-waste) deriving from the post-harvest treatment of cereals and legumes as the growth substrate for selected biosurfactant-producing microbial strains. The agricultural by-products were easily metabolized and highly effective for the growth of microorganisms and the production of rhamnolipids and surfactin by *Pseudomonas aeruginosa* and *Bacillus subtilis*, respectively. In particular, the use of corn chaff ("bee-wings") was suitable for the production of rhamnolipids. Indeed, in corn-chaff-based media, rhamnolipids yields ranged from 2 to 18 g/L of fermentation broth. This study demonstrated that the use of waste raw materials could be applied to reduce the carbon footprint of the production of biosurfactants without compromising the possibility of having a suitable fermentation medium for industrial production.

**Keywords:** rhamnolipid; surfactin; *Pseudomonas*; *Bacillus*; biosurfactants; agro-waste; corn chaff

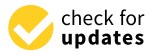

## 1. Introduction

The production chains of the agricultural sector generate abundant by-products and organic waste (herein also referred to as "agro-waste" or "waste agricultural biomass"). The quantities of the above are often evaluated only marginally by the official statistics of the sector, causing underestimations and inconsistent data. This translates into an unclear and unexhaustive cognitive framework. The latest available data on the production of vegetable waste of agricultural origin in Italy date to 1997. In 1997, the Italian production exceeded 20 million tons per year of dry matter, of which the majority (approximately 13 million tons per year) came from cereal production. This corresponds to 2–4% of the product obtained (a percentage that, in the case of the presence of mycotoxins, can grow up to 30%), which are mainly sent to anaerobic digestion plants to produce biogas [1,2]. Due to the composition, mainly consisting of cellulose, hemicellulose, and lignin [3], agro-waste is a suitable substrate for microorganism growth and the production of their metabolites [4,5]. To convert agro-waste into substrates for microbial fermentation, different processes were developed (pre-treatment). The aim of the pre-treatment is to break down

the lignocellulosic complex and increase the bioavailability of sugars (and, eventually, of other relevant nutrients) present in cellulose and hemicellulose [6]. The pre-treatments can be grouped as mechanical, chemical, and biological.

The mechanical pre-treatment of the agro-waste, which includes cutting, milling, chipping, grinding, and mixing, and, eventually, the use of ultrasounds, is applied to degrade the macro-structure of the lignocellulosic material. This increases the surface/volume ratio and exposes polysaccharides, which are easily degraded by microorganisms. To a limited extent, microwaves are also used to reduce the compactness of the lignocellulosic material to make it more accessible to enzyme action [7].

Chemical pre-treatment methods include acid and alkali hydrolysis, oxidation, and solvent extraction. The use of acids, such as $HCl$, $H_2SO_4$, $HNO_3$, $Na_2CO_3$, and $CH_3COOH$, involves the breaking of the glycosidic bonds of lignin, cellulose, and hemicellulose, which are thus converted into fermentable sugars. Alkali hydrolysis, which uses $NaOH$ or $Na_2Co_3$, is applied to liquefy lignin and a portion of hemicellulose. Alkali treatment also reduces the crystal form of cellulose, promoting the bioconversion efficiency of biomass by microorganisms [8]. The thermic pre-treatment, often associated with chemicals, employs high temperature (which is explicitly included in any axenic fermentation process), degrades lignin, and exposes cellulose and hemicellulose. The thermic pre-treatment also releases sugar monomers such as xylose, arabinose, and mannose.

Finally, biological pre-treatments using enzymes or their producing microorganisms are increasingly applied. Microorganisms can effectively produce hydrolytic enzymes. It is worth noting that the use of bacteria, such as *Bacillus* sp. or white rot fungi, can depolymerize ligninolytic structures and make the hemicelluloses more accessible and less crystalline. The biological pre-treatments are more eco-friendly and less energy-intensive and do not require the use of expensive machinery, which could have an impact on energy and infrastructure costs [9].

After the pre-treatment, the agro-wastes (in the form of liquid solutions of nutrients and/or solid residues) can be eventually combined with other nutrients, which are to be used as substrates for microbial growth and the production of microbial products [10,11].

Among the molecules which can be conveniently produced by the microbial fermentation of agro-waste, biosurfactants (BSs) and bioemulsifiers (BEs) are worth noting [12–14]. Microbial-derived BSs and BEs can act as surface-active compounds (SACs). These compounds possess amphiphilic properties, characterized by their distinct hydrophobic and hydrophilic regions, which facilitate the emulsification and dispersion of hydrophobic substances in a hydrophilic environment. BSs are known for their excellent surface activity, which involves lowering the surface and interfacial tension (ST) between different phases (liquid–air, liquid–liquid, and liquid–solid), and they possess a low critical micelle concentration (CMC), which allows the formation of stable emulsions.

Structurally, biosurfactants are highly diverse and can be classified as glycolipids, lipopeptides, phospholipids, lipopolysaccharides, fatty acids, and polymers. They find applications in cosmetics, personal care products, and household cleanings, while their potential in pharmaceuticals, environmental clean-up, agriculture, and food industries is also being explored. Glycopeptides and lipopeptides are highly represented in commercial products and are the most studied [15].

BEs are higher than BSs in molecular weight as they are complex mixtures of heteropolysaccharides, lipopolysaccharides, lipoproteins, and proteins. BEs are also known as high-molecular-weight biopolymers or exopolysaccharides. Similarly to BSs, BEs can efficiently emulsify two immiscible liquids, such as hydrocarbons (or other hydrophobic substrates) and water, but are less effective than BSs at ST reduction. Therefore, BEs are generally described as possessing emulsifying activity without surface activity [16].

BSs and BEs have several advantages over the chemical surfactants, such as a lower toxicity, higher biodegradability, better environmental compatibility, and higher selectivity and specificity at extreme temperature, pH, and salinity [17,18]. BSs and BEs of microbial origin are produced by genera such as *Bacillus*, *Pseudomonas*, and *Candida*, that produce low-molecular-weight SAC, and *Chromobacter*, *Mucor*, and *Acinetobacter*, that produce high-molecular-weight SAC [19,20]. At present, members of the genera *Pseudomonas*, *Bacillus*, *Rhodococcus*, and *Candida* are the most widely used in the industrial production of these biomolecules [15,21].

Most BSs are low-molecular-weight microbial amphiphilic molecules. They typically consist of a hydrophilic head and a hydrophobic tail and can include different subunits such as sugars, fatty acids, amino acids, or carboxylic acid groups [22]. *Pseudomonas* and *Bacillus* are very well-known surfactant producers [23]. *Pseudomonas* strains have been reported as efficient producers of glycolipids rhamnolipids, that are the most intensively studied biosurfactants [24,25]. *Bacillus* bacteria are instead known to mainly produce the lipopeptide biosurfactant surfactin.

Rhamnolipids contain one or two rhamnose moieties (mono-rhamnolipids or di-rhamnolipids) linked to β-hydroxy fatty acid chains that vary in number, length, and degree of unsaturation [26]. Approximately sixty rhamnolipids congeners and homologues have been described in the scientific literature. The predominant rhamnolipid species and the relative concentrations of the congeners are dependent on the rhamnolipid-producing strains [27,28]. Rhamnolipids are used in different industries, mostly in the petrochemical sector, but also for the bioremediation of different pollutants, in agricultural chemicals, and in personal care products. In addition, mostly due to their anti-microbial activity and their low human and environmental toxicity [27,29], they are also being considered for pharmaceutical applications [30].

Surfactin is the best-known lipopeptide biosurfactant produced by bacteria of the *Bacillus* genus. It is a cyclic lipopeptide, composed of a heptapeptide attached to a β-hydroxy fatty acid chain forming a lactone ring structure. The seven aminoacids are in the following sequence: L-Glu1-L-Leu2-D-Leu3-L-Val4-L-Asp5-D-Leu6-L-Leu7. Surfactin is mainly produced by *Bacillus subtilis*, *Bacillus amyloliquefaciens*, and *Bacillus licheniformis*. *B. subitlis* produces natural surfactins as a mixture of isoforms designated as A, B, C, and D, which have different physiological properties. They contain a minimum of eight depsipeptides with thirteen to sixteen carbons as part of their ring system. Surfactin is known to be the most powerful biosurfactant discovered so far. It displays strong emulsifying and foaming activities [31], lowering the surface tension of water from 72 to 27 mN m$^{-1}$, at a critical micelle concentration (CMC) of 20 mg L$^{-1}$.

Among BEs, the most relevant is emulsan. It is a lipopolysaccaharide bioemulsifier with a molecular weight of 1000 kDa produced by *Acinetobacter calcoaceticus* RAG-1. Emulsan is one of the most widely studied emulsifiers produced by bacteria [32]. In its pure form, emulsan shows emulsifying activity at low concentrations (0.01–0.001%). It increases the bioavailability of poorly soluble substrates in aqueous environments for microbial access and degradation by coating the hydrophobic substrate to form minicapsules. The producing bacterium can also have direct access to hydrophobic substrates, but the emulsifying activity is exhibited by the secreted emulsan.

Drawbacks in the large-scale production of microbial SAC can be the low production yield, the expensive recovery and purification, and the high costs of the fermentation substrates. However, the large range of applications (from medical and cosmetics, to the recovery of crude oil and bioremediation) of SACs of microbial origin [15,33] enhances the possibility of choosing different kinds of raw materials for fermentation depending on the value of the final products and their uses. For example, for bioremediations, the

use of standardized raw materials in the production of BS is not necessary, leading to a lower cost. Even if the BS market is still not as cost-competitive as the market of synthetic surfactants, the use of agricultural wastes in fermentation is a promising strategy to reduce the manufacturing costs [34]. The focus of this study was the production of microbial-biosurfactants using agro-waste as the main component of the fermentation media. The strategy was to match the biosurfactant-producing microorganisms with their preferred agro-waste to maximize productivity. For example, rhamnolipids and surfactin were obtained in high yields by using corn chaff and emmer/oat spelts, respectively. Most important was the fact that no pre-treatment of the agro-waste was required to reach a high productivity of BS [35,36].

This study was realized within the framework of the collaborative project "Rifiuti cerealicoli per il biorisanamento", with the acronym "RICREA" (https://www.progetto-ricrea.org/ accessed on 19 January 2025). The RICREA project was funded by the Italian Ministry for the Environment. The goal of the RICREA project was to evaluate the possibility of recovering and valorizing wastes and scraps from the production and processing of cereals and legumes. The intent was to use these wastes as substrates to produce biosurfactants which, in turn, could be used for the bioremediation of soil contaminated by hydrocarbons.

## 2. Materials and Methods

### 2.1. Microbial Strains, Culture Media, and Culture Conditions

The microorganisms used in this study were supplied by the company Madep SA and have been isolated from soils contaminated by hydrocarbons (Table 1). The isolated strains were able to produce biosurfactants at industrially interesting levels in standard media for fermentation.

**Table 1.** Microorganism supplied by Madep SA used in this study.

| Genus | Species | ID | Reference Cultural Medium | Application of the Strain |
|---|---|---|---|---|
| *Acinetobacter* | sp. | MAD90 | BCS333 | Emulsan production |
| *Bacillus* | *subtilis* | MAD3 | BCS340 | Surfactin production |
| *Rhodococcus* | *erythropolis* | MAD02B | BCS346 BCS333 BCS342 | Bioremediation of hydrocarbons and accumulation of cesium isotopes Triacylglycerole biosynthesis Biotransformation of acrylonitrile into acrylammide PHA synthesis Hydrocarbons biotransformation |
| *Candida* | *bombicola* | MADS | BCS343 | Production of sophorolipids |
| *Pseudomonas* | *aeruginosa* | MAD10 | BCS340 | Production of rhamnolipids |

The media used for the growth of the microbial strains and for biosurfactant production were obtained from BioC-CheM Solutions proprietary media database (BCSMedDat, Gerenzano (VA), Italy). All the media were prepared in 500 mL baffled flasks, and, for each flask, a volume of 100 mL was dispensed. The media were sterilized at 121–123 °C for 20–25 min. The pH of the media was measured before and after sterilization by use of a pH meter MP 120 (Mettler-Toledo GmbH, Schwerzenbach, Switzerland).

To perform viable cell count and revitalize the microbial strains, Luria–Bertani (LB) agar medium, Malt Extract (ME) agar medium, and Nutrient-Broth (NB) agar medium were used. For LB agar medium, 25 g of LB powder (10 g yeast extract; 10 g sodium chloride; and 5 g tryptone) (Becton, Dickinson and Company, Sparks, MD, USA) and 18 g of agar

(HiMedia Laboratories GmbH, Modautal, Germany) were dissolved in 1 L of ultrapure water and sterilized at 121 °C for 15 min. The pH post sterilization was 7.00 ± 0.1. For ME agar medium, 20 g of malt extract powder (17 g malt extract; 3 g peptone) (Merck KGaA, Darmstadt, Germany) and 18 g of agar (HiMedia Laboratories GmbH, Modautal, Germany) were dissolved in 1 L of ultrapure water and sterilized at 121 °C for 15 min. The pH post sterilization was 5.6 ± 0.2. For NB agar medium, 8 g of NB powder (3 g beef extract; and 5 g peptone) (Becton, Dickinson and Company, Sparks, MD, USA) and 18 g of agar (HiMedia Laboratories GmbH, Modautal, Germany) were dissolved in 1 L of ultrapure water and sterilized at 121 °C for 15 min. The pH post sterilization was 6.8 ± 0.2. All agar media prepared were poured in Petri dishes before use. The seed (vegetative) phase of growth for the bacterial strains was carried out in LB or NB broth. Then, 25 g of LB powder (Becton, Dickinson and Company, Sparks, MD, USA) (10 g yeast extract; 10 g sodium chloride; and 5 g tryptone) were dissolved in 1 L of ultrapure water and sterilized at 121 °C for 15 min. The pH post sterilization was 7.00 ± 0.1. After sterilization, 100 mL of LB broth were dispensed into sterile 500 mL baffled flasks. To control the formation of foam during fermentation, 50 μL of sterile antifoam O-10, (Merck KGaA, Darmstadt, Germany) were added to each flask. The seed (vegetative) phase of growth for the *C. bombicola* yeast was carried out in BMGY medium (10 g $L^{-1}$ yeast extract, 20 g $L^{-1}$ peptone, 10 g $L^{-1}$ glycerol, 400 μg $L^{-1}$ biotin, and 0.1 M potassium phosphate buffer at pH 6.0) [37].

The fermentation (production) media used for the production of biosurfactants are detailed in the experimental section. The media used were mainly composed of glycerol, glucose, or soybean oil as the carbon source, and yeast extract or sodium nitrate ($NaNO_3$) (Carlo Erba Reagents Srl, Cornaredo, Italy) as the nitrogen source. Potassium dihydrogen phosphate ($KH_2PO_4$) (Carlo Erba Reagents Srl, Cornaredo, Italy) was added as potassium source or to control the pH, when required. Each production medium (except the control) was also supplemented with different agro-wastes (Figure S1) which were finely ground (through the use of a cereal mill equipped with a 40-mesh sieve before use (Figure S2)). No additional treatment was applied to the agro-waste. For the corn chaff, mechanical grinding was observed to not bring any advantage and was eventually omitted.

Saline solution (NaCl 0.9%) was prepared by dissolving 9 g of sodium chloride (NaCl) (Carlo Erba Reagents Srl, Cornaredo, Italy) in 1 L of ultrapure water and sterilizing at 121 °C for 20 min. The saline solution was used for the serial dilutions for viable cell count and to suspend the microorganisms for the inoculum.

The Nutrient Glycerol solution was prepared by dissolving 20 g $L^{-1}$ of N B and 200 g $L^{-1}$ of glycerol in ultrapure water. The solution was sterilized at 121 °C for 15 min.

For the preparation of the Master Cell Banks (MCBs) and Working Cell Banks (WCBs), strains were grown on LB or NB agar solid media (for bacteria) or ME agar (for *Candida*) at 28–30 °C for 48–72 h. After the growth, the colonies were suspended in Nutrient Glycerol. The suspension was homogenized until the bacterial or yeast pellet was completely suspended. Then, 1 mL of the solution was dispensed in cryovials and stored at −80 °C.

The fermentation was performed according to the steps reported in Figure S3. One vial of the WCB was used to inoculate one agar plate which was incubated at 28 °C for 18 h. A loop of cells from the agar plate was used to inoculate 100 mL of liquid medium dispensed into a 500 mL baffled flask. The flask was then incubated at 28 °C at 200 rpm till the $OD_{600}$ (measured with a spectrophotometer UV-160, Shimadzu, Kyoto, Japan) reached a value of 2.0–2.5 (for bacteria) and 10–15 (for *Candida*). Then, 1% to 5% of the grown culture was used to inoculate 100 mL of the production medium dispensed in a 500 mL baffled flask. The production flasks were then incubated at 28 °C at 200 rpm for up to 200 hours.

To measure the growth on the production media, a viable cell count (as CFU m $L^{-1}$) was performed on serially diluted cultures plated on agar media.

*2.2. Analytical Methods*

2.2.1. Acid Hydrolysis and HPLC Quantification of Rhamnolipids and Sophorolipids (Glycolipids)

The concentration of glycolipids was routinely measured as sugar equivalents (rhamnose for rhamnolipids or glucose for sophorolipids) by conducting acid hydrolysis on the culture sample. The quantification of sugars was performed by High-Performance Liquid Chromatography (HPLC), on an Agilent technology 1260 infinity HPLC instrument (Agilent Technology, Santa Clara, CA, USA), using an isocratic HPLC method for rhamnose and glucose. The samples for analysis were prepared as described below.

First, 1 mL of sample from the fermentation broth was dispensed in a 2 mL Eppendorf tube. Then, 160 μL of 37% HCl (Carlo Erba Reagents Srl, Cornaredo, Italy) were added and the tube was shaken (HCl final concentration 5% $v\,v^{-1}$). The sample was incubated on Thermomix for 4 h at 95 °C (hydrolysis step). After hydrolysis, the sample was centrifuged for 5 min at 16,000 rcf. Next, 900 μL of the supernatant were transferred in a clean 2 mL Eppendorf tube and 100 μL of 35% $v\,v^{-1}$ perchloric acid (HClO$_4$) (Carlo Erba Reagents Srl, Cornaredo, Italy) were added. For media containing oil, 1000 μL of chloroform (CH$_3$Cl) (Carlo Erba Reagents Srl, Cornaredo, Italy) were added before centrifugation to remove the oily phase. The tube containing the supernatant was vortexed and was then placed at −20 °C for 10 min. Then, 55 μL of 7 M potassium hydroxide (KOH) (Carlo Erba Reagents Srl, Cornaredo, Italy) $w\,v^{-1}$ were added and the tube was shaken. The tube was then centrifuged at 16,000 rcf for 2 min and filtered on a 0.22 μm PES pore membrane. The filtered solid was analyzed by using the HPLC method described in Table 2.

**Table 2.** HPLC method for the quantification of rhamnose.

| Instrument: | Agilent Technologies 1260 Infinity |
|---|---|
| Column | Aminex HPX-87H (BioRad) 300 × 7.8 mm |
| Mobile Phase | 5 mM sulfuric acid |
| Flux | 0.6 mL min$^{-1}$ |
| Gradient | isocratic |
| Injection volume | 10 μL |
| Temperature | 30 °C |
| Detector | Refractive Index Detector (RID) |
| Run Time | 30 min |

The results of the HPLC analysis conducted on the samples from the hydrolyzed fermentation broth were expressed as L-rhamnose or D-glucose content. The corresponding glycolipid concentrations were calculated from the obtained L-rhamnose or D-glucose content by applying the correction factors reported by Kobayashi and co-workers [38].

2.2.2. LC-MS Analysis of Rhamnolipids

Then, 500 mg of crude fermentation broth were extracted with 500 mL of water/methanol (H$_2$O/MeOH) (50:50). After centrifugation at 16,000 rcf, the supernatant was diluted with 10 volumes of (H$_2$O/MeOH) (50:50). LC-MS analysis was performed using a 1290 Infinity Agilent Instrument (Agilent Technology, Santa Clara, CA, USA) according to the analytical method reported in Table 3.

**Table 3.** LC-MS method for the identification of rhamnolipids congeners.

| Column: | Hypersil ODS 250 × 4.6 mm, 5 µm | | |
|---|---|---|---|
| Mobile phase A | 10 mM ammonium acetate (MeCOONH$_4$) pH 7.4 | | |
| Mobile phase B | Acetonitrile (MeCN): 10 mM ammonium acetate (MeCOONH$_4$) pH 7.4 = 80:20 | | |
| Flow | 0.5 mL min$^{-1}$ | | |
| Injection Volume | 20 µL | | |
| Detector | UV (λ = 230 nm) | | |
| MS | 4000 V, negative, 200/1000 $m\ z^{-1}$, frag:VAR | | |
| Temperature: | 25 °C | | |
| | Time (min) | Mobile phase A (%) | Mobile phases B (%) |
| | 0 | 70 | 30 |
| Gradient: | 50 | 10 | 90 |
| | 55 | 10 | 90 |
| | 56 | 70 | 30 |
| | 66 | 70 | 30 |
| Stop time | 66 min | | |

### 2.2.3. HPLC Analysis of Surfactin

The samples used for the analyses were prepared from the culture broth according to the following steps. The broth was corrected at pH to 2 with 6 N HCl, 1 volume of methanol (MeOH) was added, and the sample was stirred for 10 min. After stirring, the sample was centrifuged at 16,000 rcf for 5 min and the supernatant was transferred to HPLC vials. The samples were analyzed by use of a 1260 Agilent HPLC (Agilent Technology, Santa Clara, CA, USA) according to the method described in Table 4 [39].

**Table 4.** HPLC method for the analysis of surfactin from fermentation broths.

| Column | LiCrosphere RP18 (150 × 4.6 mm, 5 µm) |
|---|---|
| Mobile Phase | Water:acetonitrile:trifluoroacetic acid 20:80:0.025% |
| Flow | 1 mL min$^{-1}$ |
| Gradient | Isocratic |
| Injection volume | 10 µL |
| Temperature | 25 °C |
| Detector | UV (λ = 205 nm) |
| Run Time | 25 min |

The surfactin concentration was calculated based on a calibration curve obtained from the use of a surfactin standard (Sigma-Aldrich, Saint Louis, MO, USA).

### 2.3. Oil Displacement Test (ODA)

The oil displacement test is a rapid and effective qualitative method to evaluate the surfactant activity of the molecule under investigation. This test, developed by Morikawa in 2000 [40], exploits the ability of biosurfactants to create circular zones (halos) in which the oil is displaced once added on top of oil deposited on top of water. The size of the displacement halo is roughly proportional to the activity and to the concentration of the biosurfactant. The test is performed in 5 cm-diameter Petri dishes, in which 30 µL of low-density crude oil are layered on top of 3 mL of ultrapure water. For the test, 3 µL of the sample containing biosurfactants are dropped on top of the crude oil layer. The diameter of the halo obtained after dropping the sample gives a qualitative indication of the surfactant efficacy. The results of this test are expressed as diameter size of the concentric

halo or simply using the (+) or (−) sign to indicate the presence and size of halos even if not concentric with respect to the Petri dish.

### 2.4. Emulsification Index ($EI_{24}$(%))

The emulsification index ($EI_{24}$(%)) [41] is a parameter used for determining the emulsifying power of a surfactant molecule. The index measurement was performed by mixing equivalent volumes (usually 2 mL) of the solution containing the biosurfactant and of *n*-hexadecane. The mixture was vortexed at high speed for 2 minutes and the measurement was made after letting the mixture at 25 °C for 24 h. The emulsification index is calculated as the ratio (expressed as percentage) between the height of the emulsified phase and the total height of the liquid column. Crude oil can be used as an alternative to *n*-hexadecane.

### 2.5. Qualitative Analysis of the Biosurfactants by Thin-Layer Chromatography (TLC)

Crude extracts of biosurfactants were qualitatively analyzed by thin-layer chromatography (TLC) on silica gel. For rhamnolipids, the TLC plate was developed using a mobile phase composed of chloroform ($CHCl_3$), methanol (MeOH), and ultrapure water ($H_2O$) in a 65:15:1 ratio. The orcinol reagent (suitable for detecting the presence of sugars, glycolipids, and glycosides) was used for the visualization. For surfactin, the TLC plate was developed using a mobile phase composed of chloroform ($CHCl_3$), methanol (MeOH), and ammonium hydroxide ($NH_4OH$ 30%) in a 65:25 + 4% ratio. The surfactin spots were visualized using ultrapure water (suitable for hydrophobic molecules).

### 2.6. pH Analysis

pH is a critical parameter in fermentation. The pH of a sample of culture broth was measured using a Mettler–Toledo pHmeter (Mettler-Toledo GmbH, Schwerzenbach, Switzerland). The pH value provides useful information on the metabolism of the microorganism during fermentation.

### 2.7. Microscopic and Macroscopic Monitoring

During fermentation, the bacterial culture was monitored both macroscopically and microscopically. Macroscopical analysis to evaluate the growth of the microorganism included turbidimetry (by measuring $OD_{600}$ on LB medium with a CE2010 spectrophotometer, Cecil Instruments, Cambridge, UK) and viscosity (on thick media containing agro-waste with a DV1 Viscometer, Brookfield, Middleboro, MA, USA). The visual observation of foam was instead used to evaluate the biosurfactant production. Microscopical analysis to monitor oil emulsification and uptake, possible contamination, biosurfactant production, and microorganism growth was conducted using a Zeiss Axioskop microscope (Carl Zeiss, Jena, Germany), or a Zeiss Stemi SV6 steremicroscope (Carl Zeiss, Jena, Germany), both equipped with an OPTIKA C-HP4 digital camera (OPTIKA, Ponteranica (BG), Italy).

### 2.8. Extraction of Rhamnolipids from Fermentation Broths

To extract rhamnolipids from the fermentation broth, the protocol from Zhang et al. [42] was used. The protocol uses $(NH_4)_2SO_4$ to make the water (fermentation broth) and a water-miscible solvent (2–propanol) immiscible. More details are reported in the experimental section.

## 3. Results

### 3.1. Chemical Composition of the Agro-Wastes

The agro-wastes used in this study were oat/emmer hull, corn chaff, and pea pod hull. The details of the origin of the agro-wastes are reported in Figure S2. These agro-wastes were chosen as they are widely available in Italy, they are low-cost, and they are

relatively easy to use in fermentation processes (after milling and sewing, or without any treatment). The composition of the agro-wastes is reported in Tables S1–S3 and was determined according to the methods reported in the "Reference or source" column of the same tables. The analysis evidenced that the agro-wastes used in this study were rich in nutrients useful for bacterial and fungal growth. The analysis also indicated that their composition is similar to the composition reported in the literature for the same materials [43,44].

*3.2. Testing of Agricultural Waste as the Growth Substrate for Target Biosurfactant Producing Microorganisms*

Samples of the culture media that allowed the growth of microorganisms were prepared by suspending 100 g of each milled agro-waste in 1 L of ultrapure water, followed by heat sterilization (123 °C for 20 min). The chemical analyses results (Tables S1–S4) showed that all the agro-wastes analyzed were rich in nitrogen and carbon and showed the presence of phosphates. Based on these results, it was determined that a cultivation medium formulated with this type of material as the sole ingredient could be suitable for microbial growth. Model microorganisms were selected among those identified as producers of biosurfactants and available from the company MADEP SA, and were grown on these media. All the selected model microorganisms demonstrated the ability to grow on the media formulated as above (colony-forming units > $10^9$) (Table 5). Based on the pH trend and the consumption of detectable carbon sources (starch and reducing sugars), we argued that different types of metabolism characterized growth on the different substrates (Figure S4).

**Table 5.** Maximum growth (CFU m $L^{-1}$) and production of biosurfactants (ODA test, indicated as + or −) achieved on media composed of each agro-waste sterilized in water. Reference values (growth and production of biosurfactants on seed and productive medium of the BioC-CheM Solutions media database and not containing agro-waste) are indicated in table (Control). The microbial inoculum was $1 \times 10^7$ CFU m $L^{-1}$ for each culture. Values reported are the average of at least 3 independent experiments with an SD below 5%.

| Strain | ID | Oat and Emmer Chaff | | Corn Chaff | | Proteic Pea Pod Hull | | Control | | Biosurfactant Produced in Control Conditions |
|---|---|---|---|---|---|---|---|---|---|---|
| *Acinetobacter* sp. | MAD90 | $1.1 \times 10^{10}$ | − | $9.0 \times 10^9$ | − | $5.0 \times 10^9$ | − | $9.0 \times 10^9$ | + | Emulsan |
| *Bacillus subtilis* | MAD3 | $3.7 \times 10^9$ | + | $1.3 \times 10^9$ | + | $7.6 \times 10^9$ | + | $5.5 \times 10^9$ | + | Surfactin |
| *Candida bombicola* | NA | $1.0 \times 10^9$ | − | $2.7 \times 10^9$ | − | $3.8 \times 10^8$ | − | $7.5 \times 10^7$ | + | Sophorolipids |
| *Pseudomonas aeruginosa* | MAD10 | $4.3 \times 10^9$ | − | $2.7 \times 10^{10}$ | + | $2.2 \times 10^{10}$ | − | $7.0 \times 10^9$ | + | Rhamnolipids |
| *Rhodococcus erythropolis* | MADO2B | $1.4 \times 10^9$ | − | $1.7 \times 10^9$ | − | $4.3 \times 10^9$ | − | $5.0 \times 10^9$ | + | Trehalolipids |

*Acinetobacter* sp. showed growths on the oat/emmer hull and on corn chaff media which are comparable to the growth observed on the control media. Good growth was also observed on the pea pod hull (Table 5). The pH trend during growth had a similar profile for all the agro-wastes, with a slight basification (higher pH values reached) on the emmer and oat chaff and on the pea pod hull. A remarkable acidification (lower pH values reached) was instead observed on the corn chaff (Figure S4A). Acidification in the presence of the corn chaff could be due to the presence of reducing sugars (3.6 g $L^{-1}$ for corn chaff vs. 1.3 g $L^{-1}$ for emmer/oat hull) and starch. Notably, most *Acinetobacter* sp. are not capable of utilizing glucose as a carbon source [45]. For those strains able to degrade glucose, the gluconolactone/gluconate pathway is used and the reaction can be readily detected by the acidification of medium in the presence of D-glucose [46].

*Bacillus subtilis* showed the best growth on the pea pod hull and emmer/oat hull, while, on the corn chaff, the CFU m $L^{-1}$ were lower and longer incubation times were required to reach the stationary phase (up to 160 h vs. 50 h for the other agro-wastes). The growth is generally affected by the surrounding environment, including the medium composition, the balance of the nutrients, and other parameters as pH. The slow growth observed on corn chaff was, therefore, probably due to suboptimal pH conditions and to the suboptimal nutrient availability. The pH trend showed a progressive increase on the emmer and oat hull and on the pea pod hull. On the corn chaff, a constant pH value around 6 was, instead, observed (Figure S4B). Poor growth on the corn chaff could also be due to the low level of free nitrogen and proteins compared to the other agro-wastes (see Tables S1–S3) [47].

*Candida bombicola* was selected for its ability to produce sophorolipids [48]. It showed good growth on the corn chaff as already reported [49]. The pH trend was consistent with the physiology of the strain: acidification was observed on the corn chaff, while, on the emmer/oat hull and on the pea pod hull, the culture pH remained stable around neutrality (Figure S4C).

*Pseudomonas aeruginosa* was chosen for its ability to produce rhamnolipids and for its nutritional versatility. The growth values obtained were above the control media on the corn chaff and pea pod hull, while, on the emmer/oat hull, a lower growth was obtained (Table 5). The pH trend was similar for all three agro-wastes evaluated, with basification and reaching pH values up to pH 9 at the end of the exponential growth phase (Figure S4D). It is worth noting that the maximum *P. aeruginosa* growth was reached at around 100 h, while, for the other strains, the lag-phase of growth was usually reached within 50 h.

*Rhodococcus erythropolis* was chosen for its versatile metabolism and its use in many soil bioremediation processes [50,51]. This strain is also capable of producing biosurfactants. Good growth was observed on the pea pod hull and corn chaff, while less abundant growth occurred on the emmer/oat hull (Table 5). The pH analysis showed a trend towards basification on the pea pod hull and on the emmer/oat hull, while, on the corn chaff, a constant pH of 6 was observed (Figure S4E).

In conclusion, the agro-wastes used in this study were a suitable growth medium for microorganisms able to produce biosurfactants. The possibility of using agro-waste biomass as a substrate for microbial growth was reported by other authors. This relies on the ability to use hemicelluloses and cellulose as a carbon and energy source, due to the presence of a suitable enzyme array [52].

### 3.3. Production of Biosurfactants from Microbial Strains Grown on Agro-Wastes

In some of the growth tests described above, the media formulated with agro-waste were also suitable for the production of biosurfactants (Table 5). The oil displacement activity test (ODA) was used to qualitatively ascertain the presence of biosurfactants in the microbial cultures. The ODA test allowed us to detect the production of surfactants in the culture broths of *Bacillus subtilis* MAD3 (Figure 1) and *Pseudomonas aeruginosa* MAD10 (Figure 2). The other strains, instead, did not show any ODA activity (both whole fermentation broth and supernatant were tested). *Bacillus subtilis* showed positive ODA results upon growth on all the agro-wastes tested, while *Pseudomonas aeruginosa* showed positive results only on the corn chaff. Negative controls were performed by dropping the supernatant from the abiotic media on plates prepared with water and crude oil. The $EI_{24}$ (%) test gave results consistent with the ODA results (Figures 1 and 2).

In a preliminary characterization, the biosurfactants were extracted in a solvent and tested by thin-layer chromatography (TLC) as described in the Section 2 (Figure 3). Figure 3A shows the comparative results between *Bacillus subtilis* samples and a surfactin

standard (Sigma-Aldrich). Figure 3B shows the results obtained on the corn chaff from *Pseudomonas aeruginosa* cultures compared with a rhamnolipid standard (Sigma-Aldrich).

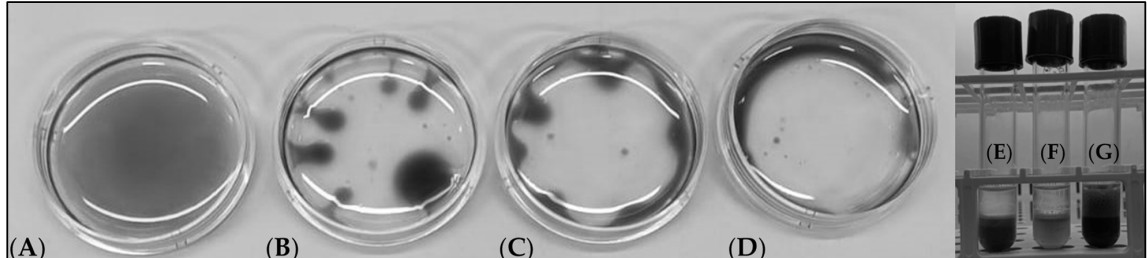

**Figure 1.** Biosurfactant tests of activity (ODA and EI$_{24}$ (%)) on crude oil in water. The supernatants of *B. subtilis* MAD3 cultures were used. From left to right: negative control obtained with the abiotic media (**A**) (similar results were obtained for all the abiotic media), oat and emmer chaff culture (**B**), corn chaff (**C**), and proteic pea pod hull (**D**). EI$_{24}$ (%) was, respectively, 10% with oat and emmer chaff culture (**E**), 10% with corn chaff culture (**F**), and 50% with proteic pea pod hull culture (**G**).

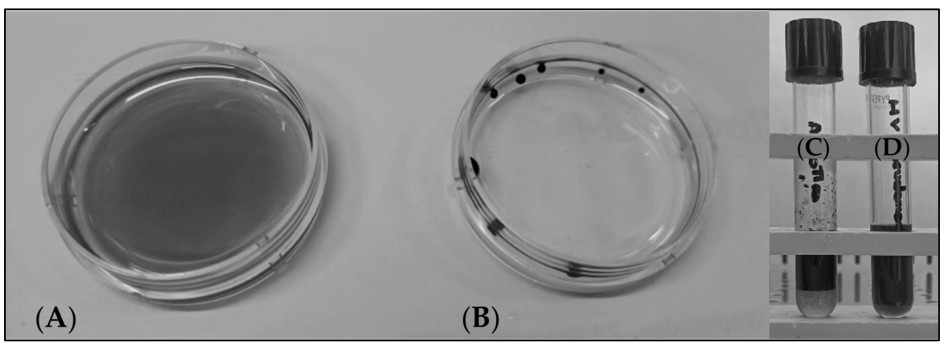

**Figure 2.** Biosurfactant tests of activity (ODA and EI$_{24}$ (%)) on crude oil in water. The supernatant of *P. aeruginosa* MAD10 cultures in medium with corn chaff were used. From left to right: ODA negative control obtained with the abiotic medium (**A**), and ODA with corn chaff culture (**B**). EI$_{24}$ (%) obtained with the abiotic medium corn chaff, 50% (**C**), and EI$_{24}$ (%) obtained with corn chaff culture, 90% (**D**). The high value of EI$_{24}$ (%) in the control (**C**) was due to the emulsifying activity of residual starch contained in the corn chaff.

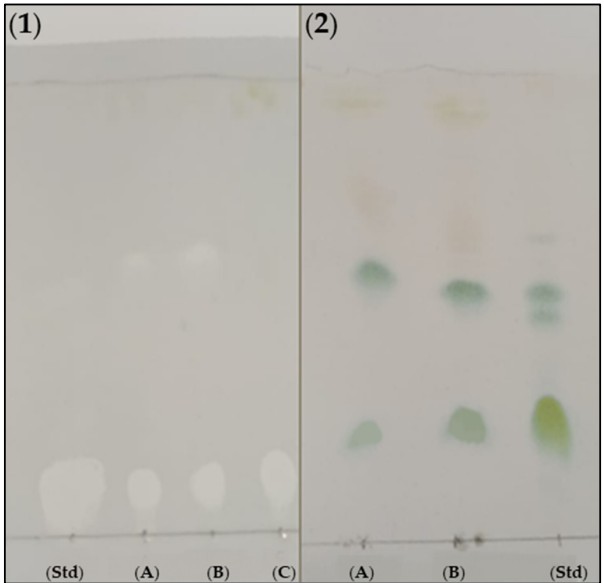

**Figure 3.** TLC of B. *subtilis* MAD3 extracts (panel (**1**)) from oat and emmer hull culture (**A**), corn chaff (**B**), pea pod hull (**C**), and *P. aeruginosa* MAD10 extracts from corn chaff cultures (panel (**2**), (**A**,**B**)). Standards of surfactin (panel (**1**), (**Std**)) and rhamnolipids (panel (**2**), (**Std**)) were used as controls.

The quantitative determination of surfactin from *B. subtilis* MAD3 and of rhamnolipids from *P. aeruginosa* MAD10 was performed by HPLC as described in the Section 2. The productivity for rhamnolipids is reported in Table 6.

**Table 6.** Formulation of corn-chaff-based media for the production of rhamnolipids by *P. aeruginosa* MAD10. Values reported are the average of at least 3 independent experiments with an SD below 5%.

| Trial ID# | Components of the Fermentation Medium | Amount for Each Component (g L$^{-1}$) | Maximum Rhamnolipids Production Achieved (g L$^{-1}$) |
|---|---|---|---|
| A | Corn Chaff<br>Glycerol<br>NaNO$_3$<br>KH$_2$PO$_4$ | 100<br>40<br>2<br>1 | 11.8 |
| B | Corn Chaff<br>Glycerol<br>KH$_2$PO$_4$ | 100<br>40<br>1 | 9.4 |
| C | Corn Chaff<br>Glycerol<br>Soybean Oil<br>NaNO$_3$<br>KH$_2$PO$_4$ | 100<br>40<br>20<br>2<br>1 | 17.9 |
| D | Corn Chaff<br>Glycerol<br>WCO<br>NaNO$_3$<br>KH$_2$PO$_4$ | 100<br>40<br>20<br>2<br>1 | 16.4 |
| E | Corn Chaff<br>Glycerol<br>NaNO$_3$<br>KH$_2$PO$_4$ | 100<br>60<br>2<br>1 | 11.1 |
| F | Corn Chaff<br>Soybean Oil<br>NaNO$_3$<br>KH$_2$PO$_4$ | 100<br>20<br>2<br>1 | 8.1 |
| G * | Glycerol<br>NaNO$_3$<br>KH$_2$PO$_4$ | 40<br>2<br>1 | 0.0 *** |
| H | Corn Chaff<br>NaNO$_3$<br>KH$_2$PO$_4$ | 100<br>2<br>1 | <2.0 ** |
| I * | Soybean Oil<br>NaNO$_3$<br>KH$_2$PO$_4$ | 20<br>2<br>1 | 0.0 *** |
| J | Oat and Emmer Hull<br>Corn Chaff | 50<br>50 | <2.0 ** |
| K * | Oat and Emmer Hull<br>Pea pod hull | 50<br>50 | 0.0 *** |
| L | Corn Chaff | 100 | <2.0 ** |
| M | Corn Chaff<br>Soybean Oil | 100<br>20 | 8.6 |
| N | Corn Chaff<br>WCO | 100<br>20 | 6.1 |

**Table 6.** *Cont.*

| Trial ID# | Components of the Fermentation Medium | Amount for Each Component (g L$^{-1}$) | Maximum Rhamnolipids Production Achieved (g L$^{-1}$) |
|---|---|---|---|
| O | BCS340 (positive control, Industrial Medium) | | 15.0 |
| | Glucose | 20 | |
| | Glycerol | 40 | |
| | Soybean oil | 20 | |
| | Soybean meal | 20 | |

* negative control; ** less than 0.3–0.4 g L$^{-1}$ of rhamnose equivalent to 1–2 g L$^{-1}$ of RLs, trace visible by TLC; *** no trace by TLC; WCO: waste cooking oil.

### 3.4. Formulation of a Suitable Fermentation Medium Based on Corn Chaff for the Production of Rhamnolipids

As reported previously, *P. aeruginosa* MAD10 was capable of producing rhamnolipids when grown on media formulated with corn chaff and water and displayed a fast and abundant growth. In cultures of *P. aeruginosa* MAD10, the oil displacement activity (ODA) was also tested with samples of the broth culture from mixtures of different agro-wastes. The only mixture that gave a positive ODA result was that with the oat/emmer hull plus corn chaff, while negative results were obtained with the oat/emmer hull plus pea peel, confirming the data obtained with the individual agricultural waste and the peculiarity of the corn chaff in stimulating the biosurfactant production (Table 6). As corn (*Zea mays* L.) is one of the most produced cereals worldwide and the by-products of corn cultivation are estimated to be approximately $1.64 \times 10^8$ tons globally, this result is particularly interesting for the development of a large-scale production process based on this agro-waste. For this reason and considering the high productivity in rhamnolipids (Table 6), we concentrated our efforts on corn chaff as the fermentation medium nutrient.

The corn chaff is mainly composed of cellulose and hemicellulose (approximately 75%), lignin (approximately 20%), starch (approximately 0.3%), and proteins (approximately 2.3%) [53,54]. The sugar composition of hemicellulose is mainly arabinose (16.4%), galactose (5.3%), xylose (75.7%), and glucuronic acid (1.9%). Corn chaff is abundant (21% of the corn waste) [53], is easy to degrade due to its delicate structure (described as bee-wings), and can be used "as is" in the formulation of the fermentation media. In contrast, the other agro-wastes evaluated this study had to be ground and sewn before being introduced in the fermentation medium. Due to the interesting characteristics of corn chaff and its ability to sustain the rhamnolipids production, we investigated it further by designing an improved fermentation medium. Mixtures composed of corn chaff and standard nutrients used in fermentation were tested. The combinations of nutrients and the production of rhamnolipids are reported in Table 6. The quantification of rhamnolipids was performed as described in the Section 2.

A titration of the rhamnolipids present in the culture broth at the time of harvest (approximately 144 h for the industrial medium) was performed by HPLC. The best results were obtained with trial C (below also identified as medium BCS388), and trial D (Table 6). Trial C gave 18 g L$^{-1}$ of rhamnolipids and Trial D gave 16 g L$^{-1}$ (vs. a maximum of 2 g L$^{-1}$ with media formulated with the corn chaff and inorganic salts only and a maximum of 2 g L$^{-1}$ with the corn chaff in water) (Table 6). The results of those tests indicated that vegetable oil (both soybean oil and exhausted sunflower oil, a waste cooking oil of considerable interest from a circular economy perspective) and glycerol play an important role in the maximization of the rhamnolipid production when combined with inorganic salts and corn chaff. On the other side, glycerol or soybean oil, when combined with inorganic

nitrogen and phosphorous, are not enough to warrant the production of rhamnolipids (Trials G and I). The rhamnolipid productivity obtained is among the highest productivities reported in the literature for batch fermentations and for wild-type strains [24], indicating that corn chaff is an excellent substrate.

The mono-rhamnolipid and di-rhamnolipid percentages were determined by LC-MS (Table 4). The mono- and di- rhamnolipids ratio was 5:95 in all analyses. This result suggested that the ratio between mono-/di-rhamnolipids is strain-specific rather than fermentation-medium-specific, as already reported in the literature [28].

### 3.5. Evaluation of the Optimal Corn Chaff Concentration and of the Effect of α-Amylase on Rhamnolipid Production

A drawback of the fermentation media formulated with the corn chaff was the high viscosity post-sterilization, which limited the corn chaff concentration in the fermentation medium to 100 g $L^{-1}$. With a corn chaff concentration above 100 g $L^{-1}$, the medium reached a jelly consistence, which was not suitable for liquid-phase fermentations. To reduce the viscosity of the broth and to verify the optimal corn chaff concentration for the rhamnolipids production, trials were conducted with decreasing concentrations of corn chaff. BCS388 media with 25 g $L^{-1}$, 50 g $L^{-1}$ and 75 g $L^{-1}$ of corn chaff were prepared. The resulting fermentations were compared with the BCS388 medium with 100 g $L^{-1}$ of corn chaff (reference medium), which was considered the optimal medium for rhamnolipid production, as shown in Figure 4 (higher corn chaff concentrations could not be tested due to excessive viscosity). After the revitalization of the strain and initial growth in the vegetative media, 1% of the grown vegetative culture was inoculated in 500 mL baffled flasks containing 100 mL of medium, as described in the Section 2. All media formulated with a corn chaff content lower than in the reference medium displayed a consistent decrease in viscosity (170, 37, and 2.5 cP for the 75, 50, and 25 g $L^{-1}$ corn chaff, respectively). However, these media also showed rhamnolipid productivities lower than that in reference medium, specifically, a reduction in productivity of approximately 23% (for the 75 g $L^{-1}$), 13% (for the 50 g $L^{-1}$), and 41% (for the 25 g $L^{-1}$). Among the three media, the one with the higher productivity is the one with 50 g $L^{-1}$ of corn chaff. This could be the result of a compromise between the nutrient availability (including oxygen availability) and viscosity.

As the high viscosity of the broth is a challenge especially for large-scale fermentations, this issue was investigated further by treating the BCS388 medium with α-amylase. The hypothesis was that the starch and/or other polysaccharides with α-linked D-glucose units released during the sterilization of the medium could contribute to the viscosity. Therefore, to reduce the viscosity of the broth, α-amylase was used under controlled conditions to hydrolyze the polysaccharides being released. The α-amylase was added to the fermentation broth and, after hydrolysis, was heat-inactivated during the medium sterilization. The reference BCS388 medium (containing 100 g $L^{-1}$ of corn chaff) was used for these experiments. The concentrations of α-amylase equivalent to 1.25 U $mL^{-1}$, 2.5 U $mL^{-1}$, and 5 U $mL^{-1}$ were added to the reference BSC388 medium as described in the Section 2. After the strain revitalization and initial growth in vegetative media, 1% of the vegetative culture was then inoculated in 500 mL flasks containing 100 mL of media treated with different α-amylase contents, as described in the Section 2.

The treatment with α-amylase at all tested concentrations compromised the rhamnolipid production. The addition of 1.25 U $mL^{-1}$ of α-amylase reduced the viscosity from approximately 2850 cP to approximately 20 cP and the rhamnolipid production decreased by 20%. Larger quantities of α-amylase proportionally decreased the rhamnolipid production (Figure 5). The inhibition of the rhamnolipid production could be attributed to the high glucose concentration originating from the hydrolysis of starch and/or of other polysaccharides containing α-linked D-glucose units. Previous studies conducted under

our experimental conditions demonstrated that glucose has an inhibitory effect on the rhamnolipid production by *Pseudomonas aeruginosa* MAD10, contrary to what was suggested by many industrial processes where glucose was routinely used in the production of rhamnolipids.

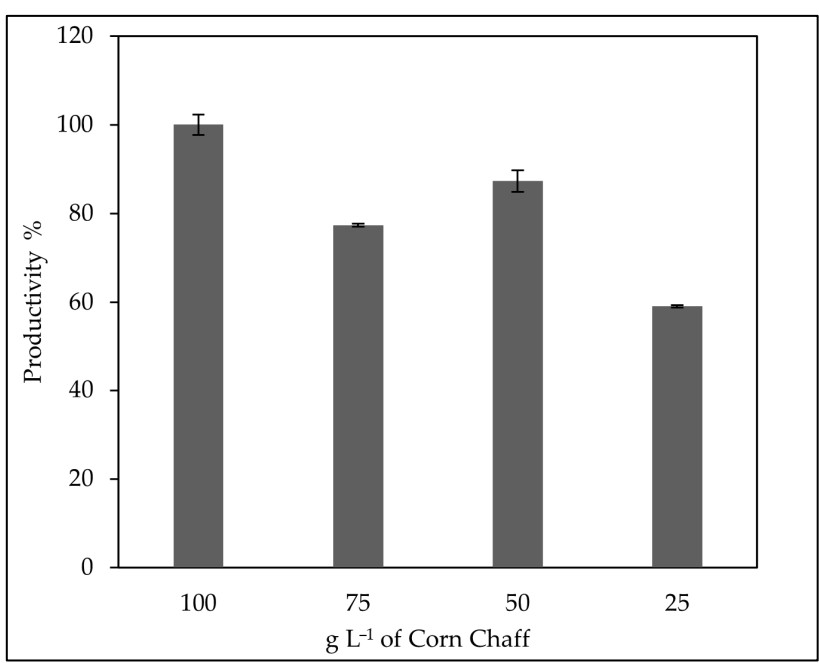

**Figure 4.** Effect of corn chaff concentration on RL production by *P. aeruginosa* MAD10. The productivity % on the y-axis is the % productivity relative to the productivity obtained with the reference medium (BSC388 containing 100 g $L^{-1}$ of corn chaff) (approximately 18 g $L^{-1}$). On the x-axis is reported the corn chaff concentration (in g $L^{-1}$).

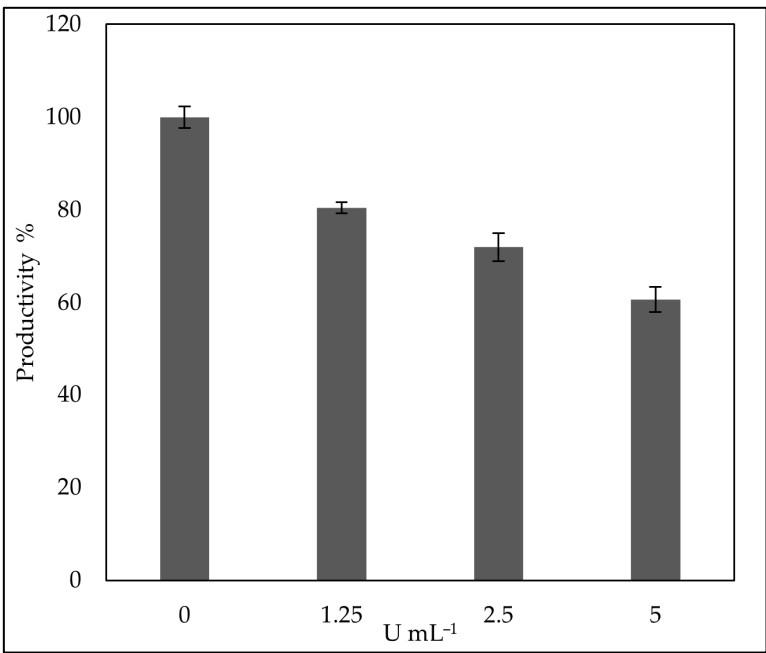

**Figure 5.** Effect of α-amylase on RL production by *P. aeruginosa* MAD10. The productivity % on the y-axis is the % productivity relative to the productivity obtained with the reference medium (untreated BSC388 containing 100 g $L^{-1}$ of corn chaff). On the x-axis is reported the α-amylase concentration (in U $mL^{-1}$).

### 3.6. Study of the Fermentation of P. aeruginosa in Medium BCS388

Glycerol is a fundamental nutrient in the production of biosurfactants and, specifically, of rhamnolipids [55]. It mainly functions as an osmoprotectant: the cell cultures grown in the presence of glycerol show a cellular physiology less subjected to stress when compared to cultures grown in the presence of only sugars as a carbon source [56]. To better understand the physiology of production in the newly formulated medium, the consumption of glycerol was monitored through HPLC as described in the Section 2. As shown in Figure 6, a correlation can be observed between the onset of the rhamnolipid production and the quantity of glycerol present in the culture medium. The increase in production of rhamnolipids ends with the exhaustion of glycerol.

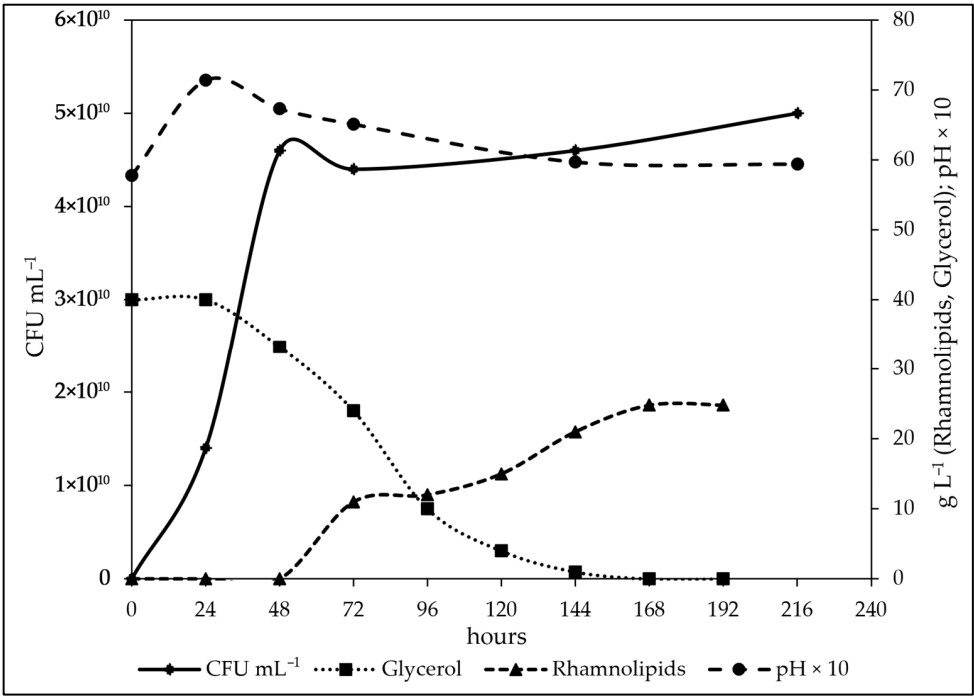

**Figure 6.** Correlation between glycerol and CFU mL$^{-1}$ and production of RLs by *P. aeruginosa* MAD10. Values reported are the average of at least 3 independent experiments with an SD below 5%.

### 3.7. Purification of Rhamnolipids from Medium BCS388 and Identification of the Different Congeners

Rhamnolipid extraction and purification trials were carried out to evaluate the effect of the presence of corn chaff on the purification of rhamnolipids, and to determine if an eco-friendly method of extraction could be applied. Currently, rhamnolipids are extracted from the fermentation broth using organic solvents (such as ethylacetate, chloroform, and dichloromethane) which are risky to handle, aggressive to the environment, and expensive to dispose [24,57]. The goal of these trials was to identify an efficient method to extract rhamnolipids using environmentally friendly solvents. An alcohol-inorganic salt system was used for the extraction/purification. 2-propanol was selected as the alcohol and $(NH_4)_2SO_4$ was selected as the salt. The addition of the salt facilitates the localization of rhamnolipids in the organic phase. The mechanism involved is the salting-out effect already described in literature [42] and the purification scheme is reported in Figure 7. The salting-out effect makes immiscible the two phases (water phase and 2-propanol phase), which are normally miscible. This facilitates the separation of the two phases, with the rhamnolipids being present in the 2-propanol phase. The rhamnolipid concentration at the various steps of the purification process was monitored by HPLC, as described in the Section 2. Table 7 shows the recovery of rhamnolipids at the different steps of purification.

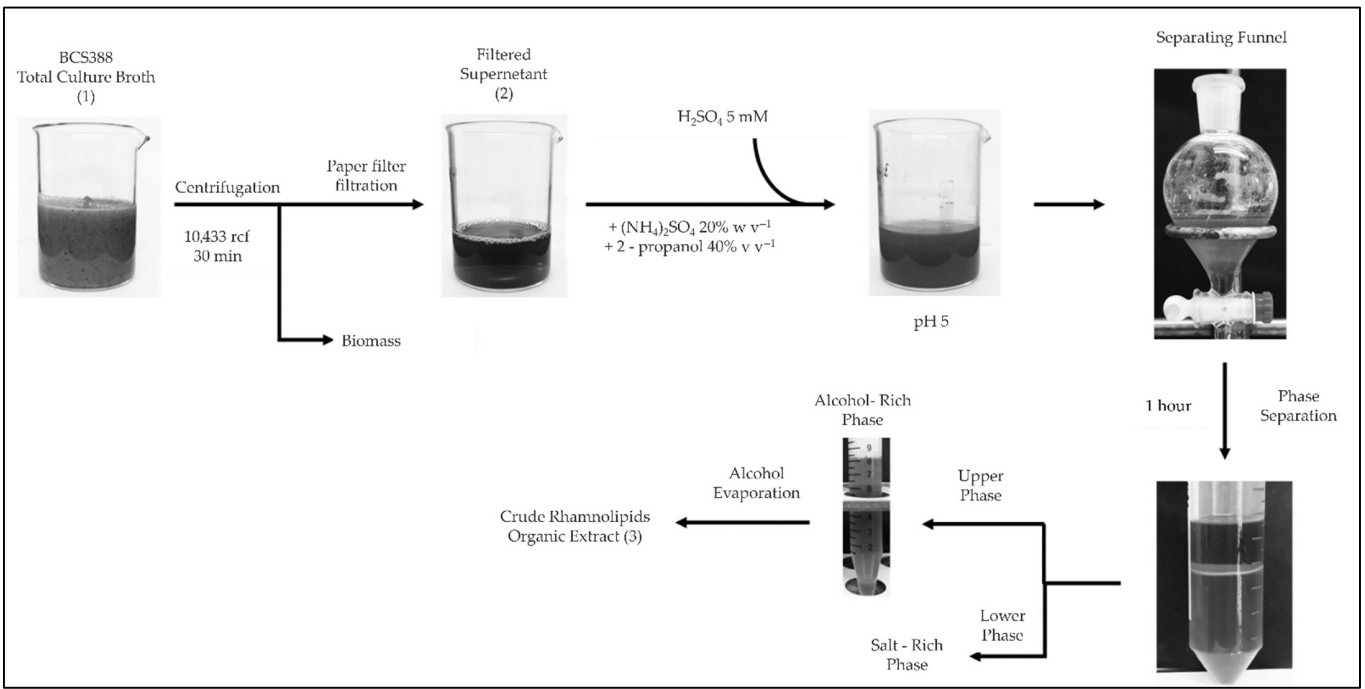

**Figure 7.** Rhamnolipid extraction/purification via aqueous two-phase system. Details in text.

**Table 7.** RL recovery in the different downstream steps. Values reported are the average of at least 3 independent experiments with an SD below 5%.

| Sample | pH | Concentration | Yield |
|---|---|---|---|
| | | g L$^{-1}$ | % |
| Total culture broth (1) | 6.2 | 12.5 | 100 |
| Filtered supernatant (2) | 6.2 | | 92 |
| Organic extract (3) | | | 63.4 |

After the first separation through the centrifugation and filter paper filtration of the culture broth, 92% of rhamnolipids were separated in the supernatant. The crude rhamnolipid organic extract appeared as a viscous oil with a content in pure rhamnolipids of 28%. Overall, the process yield was 63.4% of the initial amount of rhamnolipids in the fermentation broth. The encouraging results, especially considering the complexity of the corn chaff matrix, supported the initiation of more studies on this DSP method to improve the yield and purity.

Rhamnolipids are produced by *P. aeruginosa* as a group of related molecules (congeners or complex). The semi-purified sample described above was used for the identification of the different congeners. The analysis was performed by LC-MS (Figure 8) according to the method described in the Section 2 and the different congeners identified are reported in Table 8. The results showed that the congener composition was equivalent to the one obtained in different media during our studies with the same strain. This suggests that the media and the fermentation conditions used do not impact the congener composition, which is instead controlled by the strain utilized [28].

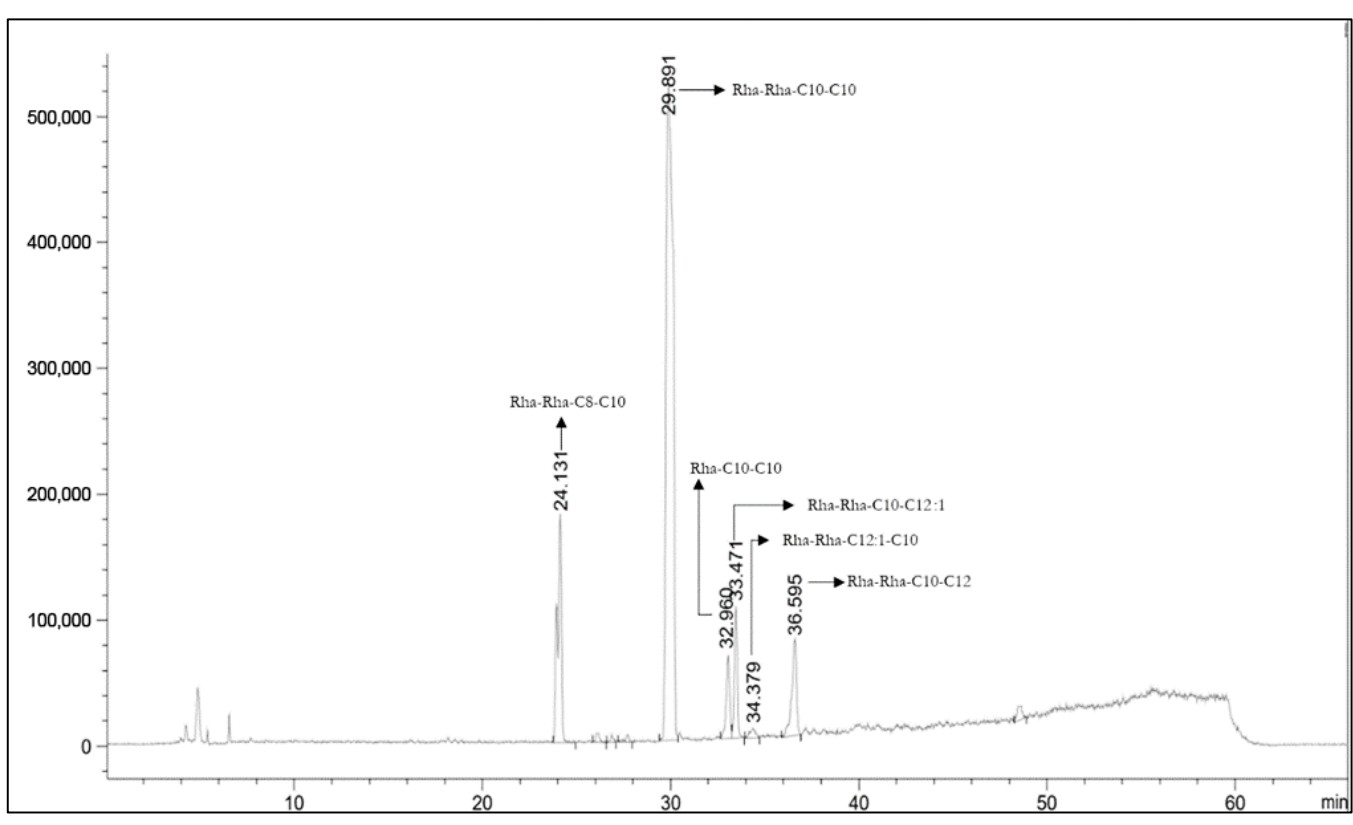

**Figure 8.** LC-MS analysis chromatogram of a typical sample of rhamnolipids.

**Table 8.** Identification of the different rhamnolipids congeners produced by fermentation in medium BCS388 from strain *P. aeruginosa* MAD10.

| Rt (min) | Compound | Structure | Area % |
|---|---|---|---|
| 24.13 | Rha-Rha-C8-C10<br>Rha-Rha-C10-C8 | | 13.44 |
| 27.53 | Rha-Rha-C10-C10 | | 66.87 |
| 30.26 | Rha-C10-C10 | | 4.18 |

**Table 8.** *Cont.*

| Rt (min) | Compound | Structure | Area % |
|---|---|---|---|
| 30.82 | Rha-Rha C10-C12:1 |  | 13.1 |
| 31.67 | Rha-Rha-C12:1-C10 |  | <1.5 |
| 3.78 | Rha-Rha-C10-C12 Rha-Rha-C12-C10 |  | <1.5 |
| | RL Tot considered | | 97.59 |

## 4. Discussion

Agricultural waste has long been known as a substrate for the growth of microorganisms and for the production of microbial biomass, metabolites, and enzymes [10,58]. The advantage of using agricultural waste in fermentation is the lower cost and the support of a zero-waste economy. However, the use of agro-waste requires costly pre-treatments, gives low production yields, and is subjected to seasonal and non-seasonal variations which affect the quality of feedstocks and, consequently, the product quality, yield, and costs [59].

In this study, we have considered the use of the widespread Italian agro-waste to produce biosurfactants. Similar studies were already reported but they all showed low production yields [13]. Our alternative approach was to couple the biosurfactant producer microorganisms with its preferred agro-waste.

The agro-wastes led to an efficient microbial growth and were also able to stimulate the production of biosurfactants in *Pseudomonas aeruginosa* and *Bacillus subtilis*. In particular, the corn chaff was found to be a suitable substrate for the growth of the *P. aeruginosa* MAD10 strain and for the production of rhamnolipids, and the oat/emmer chaff was found optimal for the growth and production of surfactin from *Bacillus subtilis* MAD3. Interestingly, *P. aeruginosa* MAD10 was able to proliferate on different agro-wastes but produced rhamnolipids only when the substrate was the corn chaff. This observation confirmed our initial hypothesis of the necessity to couple the microorganism with the preferred agro-waste. *B. subtilis* MAD3 was able to produce surfactin with all three agro-wastes but the highest yield was obtained on the oat/emmer chaff. Beside the yields in biomass and biosurfactants (particularly relevant in *P. aeruginosa* MAD10), there were

several other advantages of our approach: (i) the easy pre-treatment of the agro-waste (a milling pre-treatment which could be performed directly at the source), (ii) the use of a fermentation medium exclusively based on the agro-waste and water, and (iii) surfactant yields which are of industrial relevance (up to 18 g $L^{-1}$ in batch fermentations with a medium optimized for rhamnolipids). When comparing this yield with the one reported in the literature [60] for rhamnolipids from the corn chaff with the agro-waste treated prior to fermentation (51.6 mg $L^{-1}$), the advantages of our approach are evident.

The costs of the fermentation medium were far below those of the standard industrial media. For example, the corn chaff in Italy costs 1 € per ton, which leads to an average cost of the fermentation medium of 0.05 €/Kg of rhamnolipids (assuming a 2 g $L^{-1}$ productivity in rhamnolipids).

The reason why the corn chaff selectively supports rhamnolipid production when used as the sole fermentation substrate, and stimulates their production when used in combination with other nutrients, still remains largely unknown. We observed the negative effect of the corn chaff treatment with $\alpha$-amylase at all tested concentrations on the rhamnolipid production, indicating that free sugars are detrimental for their production. Instead, we have been able to increase the rhamnolipid yield, by adding to the medium additional carbon sources (for example, the addition of soybean oil increased the production to more than 6 g $L^{-1}$), indicating that the corn chaff was an excellent supplement for nitrogen, phosphate, and other micronutrients. Small amounts of nitrates and phosphates together with the additional carbon sources further increased the productivity towards maximum levels (18 g $L^{-1}$). It was also interesting to note that the use of media without the corn chaff but with all the other nutrients unchanged did not show any production. All of these observations suggested that the corn chaff has a balanced composition for the stimulation of rhamnolipid production or that it contains specific inducers, which are not yet identified.

## 5. Conclusions

The production of biosurfactants from agricultural waste not only adds value to organic leftovers, but it also encourages sustainable practices and reduces the environmental impact of wastes and waste disposal. To efficiently and effectively produce biosurfactants from these renewable resources, process optimization, and the selection of the optimal agricultural waste and of the microorganisms are critical. If agro-industrial wastes are used efficiently as raw materials for fermentation, the production cost of biosurfactants and, consequently, their price could significantly drop. In conclusion, we have demonstrated that agro-waste is an excellent substrate with which to produce biosurfactants. Our future studies will focus, in particular, on *P. aeruginosa*, using different strains which are known to give different rhamnolipid complex compositions, and applying a Design of Experiment (DOE) approach to improve the growth media. Our substrates are also utilized within the frame of the RICREA project for the bioremediation treatment of contaminated soils. This application is possible due to many factors, including the following: (i) the production of biosurfactants is of interest for the bioremediation of hydrocarbons, (ii) the strains used in our work are also able to degrade PAH, and (iii) cereal wastes are commonly used as soil amendments for bioremediation. The approach based on the production of biosurfactants from agricultural waste significantly shortens the time between waste generation and its reuse or recycle.

**Supplementary Materials:** The following supporting information can be downloaded at: https://www.mdpi.com/article/10.3390/fermentation11020074/s1, Figure S1: Representation and description of the crops used in this study and identification of the waste of interest in the formulation of fermentation media. Corn (A); oat and emmer (B); pea pod (C). (images from iStockphoto LP); Figure S2: Ceral mill used in the preparation of the agro-waste and different stages of preparation of

the agro-waste. Original emmer and oat chaff agro-waste (A); agro-waste after milling (B), sieving through 40 mesh (C), sieved, ready to use material (D); Figure S3: Fermentation Synoptic; Figure S4: pH determination during fermentation on agro-waste based media. Results are the average of at least three independent experiments with a SD of no more than 5%. (A) Acinetobacter sp. MAD90, (B) Bacillus subtilis MAD3, (C) Candida bombicola NA, (D) Pseudomonas aeruginosa MAD10, (E) Rhodococcus erythropolis MADO2B; Table S1: Composition of the different agro-wastes used in this study. Oat and emmer hull [61,62]; Table S2: Composition of the different agro-wastes used in this study. Corn chaff [61,62]; Table S3: Composition of the different agro-wastes used in this study. Proteic pea pod hull [62].

**Author Contributions:** Conceptualization, M.V., A.B. and F.B.; methodology, A.B. and T.B.; software, S.C.; validation, A.B., M.V. and F.B.; formal analysis, S.C.; investigation, S.C.; resources, T.B.; data curation, F.B.; writing—original draft preparation, F.B.; writing—review and editing, A.B. and S.C.; visualization, S.C.; supervision, F.B.; project administration, M.V.; funding acquisition, M.V. All authors have read and agreed to the published version of the manuscript.

**Funding:** This research was funded by MINISTERO DELL'AMBIENTE—ITALY, grant number D73C22000630001, TITLE: "Rifiuti cerealicoli per il biorisanamento (RICREA)".

**Institutional Review Board Statement:** Not applicable.

**Informed Consent Statement:** Not applicable.

**Data Availability Statement:** Data supporting the reported results can be found under www.progetto-ricrea.org (accessed on 19 January 2025).

**Acknowledgments:** We are grateful to the partners of the RICREA project—Cooperativa Quadrifoglio, Promocoop Lombardia, and Sistemi Ambientali Srl, to Carmine Capozzoli and Andrea Santoni for the LC-MS analysis, and to Paola Giaroni and Barbara Galbiati for critically revising the manuscript.

**Conflicts of Interest:** The authors Fabrizio Beltrametti, Adriana Bava, and Sara Carnelli are employed by the company "BioC-CheM Solutions S.r.l., and Trello Beffa is employed by the company "MADEP S.A.". For the purposes of this investigation, there was no financing relationship with the company; therefore, there are no conflicts of interest. The remaining authors declare that the research was conducted in the absence of any commercial or financial relationships that could be construed as a potential conflict of interest.

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
