# Peer review of "Identification of Corn Chaff as an Optimal Substrate for the Production of Rhamnolipids in Pseudomonas aeruginosa Fermentations"

_fermentation, doi:10.3390/fermentation11020074_

Round 1

Reviewer 1 Report

Comments and Suggestions for Authors The authors set the stage to investigate the use of agricultural wastes from the post-treatment of cereals and legumes, such as pea pod hall, corn chaff and oat and emmer hull, for the production of biosurfactants. For this purpose, the authors screened a few microorganisms, knowing their ability to produce biosurfactants, for their growth on the different agro-wastes. Their approach revealed that the agro-wastes supported microbial growth in all cases and were also able to support the production of biosurfactants. Corn chaff was identified as a suitable substrate for the growth of P. aeruginosa MAD10 strain and for the production of rhamnolipids, and oat and emmer chaff as supporting growth and production of surfactin by Bacillus subtilis MAD3. The P. aeruginosa MAD10 was able to grow on the different substrates, but produced rhamnolipids only when the substrate was corn chaff. The treatment of corn chaff with a amylase compromised rhamnolipids production at all tested concentrations, indicating free sugars are detrimental for production. Their study showed that production of biosurfactants from agricultural wastes encourages sustainable practices and reduces the environmental impact of waste disposal. This research work is interesting, but there are some issues that have to be addressed.   1) The authors should mention the method (e.g. equipment, software) that use to perform the microscopic and macroscopic analysis   2) The authors should describe where the signs (+) or (-) between the values are referred to ? In ODA evaluation or in production of biosurfactants ?   3) Could the authors explain why 50 g/l corn chaff has higher RLs production than that of 75 g/l corn chaff concentration (Table 4) ?  

Author Response

Dear Reviewer,

we thank you very much for critically revising our work. Below are the the answers (Italics) to your specific comments/requests (regular).

Comment 1.

1. The authors should mention the method (e.g. equipment, software) that use to perform the microscopic and macroscopic analysis  

Growth was evaluated by measuring OD600 on LB medium with a Cecil mod. CE2010 spectrophotometer and viscosity (on thick media containing agro-waste) with a Brookfield mod. DV1 Viscometer). The microscopic analysis was performed with a Zeiss Axioskop microscope equipped with an OPTIKA C-HP4 digital camera (Ponteranica, Italy), while the macroscopic analysis was performed visually or with the help of a  Zeiss Stemi SV6 stereoscope. The above information was added in the M&M section.

2. The authors should describe where the signs (+) or (-) between the values are referred to ? In ODA evaluation or in production of biosurfactants ?  

In Table 5 we generically indicate the production of biosurfactants based on positivity at the ODA test. A note was added to the table description

3. Could the authors explain why 50 g/l corn chaff has higher RLs production than that of 75 g/l corn chaff concentration (Table 4) ?  

Based on our studies, we argue that the productivity is a compromise between corn-chaff derived nutrients and viscosity (limiting oxygen distribution in the culture) of the broth. We interpret the amount of 50 g/L corn chaff as giving a correct viscosity but a limiting nutrient availability

Reviewer 2 Report

Comments and Suggestions for Authors

Dear authors, I am enclosing comments on your paper entitled Identification of corn chaff as an optimal substrate for the production of rhamnolipids in Pseudomonas aeruginosa fermentaions. 

ITEM

Comentary

1

Abstract

Although the work focuses on the use of agro-industrial wastes, the main objective seems to center on the biosurfactants produced from these carbon sources. I suggest emphasizing this aspect in the abstract.

2

Introduction

The authors conclude that good results are achieved in biosurfactant production specifically with Pseudomonas. However, upon reviewing the fermentation experiments, all of them include glycerol. This raises the question of whether it is the glycerol or the corn chaff that truly stimulates biosurfactant production. Clarifying this point would be helpful.

3

Materials and Methods

The use of Rhodococcus erythropolis is mentioned, but later it is referred to as Rhodococcus sp. It is essential to standardize the nomenclature to avoid confusion.

4

Materials and Methods

Although biosurfactant production using agricultural residues is not new, given the problems associated with these residues, questions arise regarding the preparation of the organic material by the authors. This is mentioned in the introduction but not detailed in the materials and methods section. Including this information is recommended.

5

Results

The title "Identification of microorganisms suitable for growing on agricultural waste" should be modified. Since the bacterial strains were already identified, the title should focus solely on the selection of biosurfactant-producing strains.

6

Results

In Table 6, the title should be more specific, clearly indicating that the P. aeruginosa MAD10 strain was used (if applicable).

7

Results

What is the composition of the industrial medium used? Including this information would be beneficial.

8

Results

In section "3.7. Purification of rhamnolipids from medium BCS388 and identification of the different 544 congeners," optimization of biosurfactant production is mentioned. However, I have only observed the selection of carbon sources, and no response surface assays or other experiments identifying optimal parameters are presented. This section should be revised.

9

conclusions

The conclusions should focus on the selection of bacteria capable of producing biosurfactants using only agricultural residues as sources of carbon and nitrogen. This becomes complicated when glycerol is added to the fermentation processes. I recommend improving the conclusions based on these points while highlighting the results obtained.

Author Response

Dear Reviewer,

we thank you very much for critically revise our manuscript and for your suggestions for improvement. Below are the answers (italics) to your comments/questions (regular)

ITEM

Comentary

1

Abstract

Although the work focuses on the use of agro-industrial wastes, the main objective seems to center on the biosurfactants produced from these carbon sources. I suggest emphasizing this aspect in the abstract.

We have emphasized the production of biosurfactants in the abstract according to the suggestion

2

Introduction

The authors conclude that good results are achieved in biosurfactant production specifically with Pseudomonas. However, upon reviewing the fermentation experiments, all of them include glycerol. This raises the question of whether it is the glycerol or the corn chaff that truly stimulates biosurfactant production. Clarifying this point would be helpful.

We have reported in Table 6 different media for the production of rhamnolipids. Although the best productivities are achieved with the addition of a combination of nutrients (which eventually include glycerol and soybean oil), corn chaff alone was able to support production of rhamnolipids (Trials H and L) while glycerol and soybean oil (added with nitrogen, phosphate sources and micronutrients at the concentrations we currently use for several products and strains) were not (Trials G and I). Based on that result we concluded that corn chaff was sufficient to stimulate biosurfactant production by itself while being eventually complemented by other nutrients

3

Materials and Methods

The use of Rhodococcus erythropolis is mentioned, but later it is referred to as Rhodococcus sp. It is essential to standardize the nomenclature to avoid confusion.

We have uniformed with Rhodococcus erythropolis all along the text

4

Materials and Methods

Although biosurfactant production using agricultural residues is not new, given the problems associated with these residues, questions arise regarding the preparation of the organic material by the authors. This is mentioned in the introduction but not detailed in the materials and methods section. Including this information is recommended.

We have added preparation details in the M&M section. In brief, there was no treatment besides milling and (eventually) sieving. The treatment of corn-chaff with a-amylase was performed for experimental purposed and had a negative impact (reported in the experimental section)

5

Results

The title "Identification of microorganisms suitable for growing on agricultural waste" should be modified. Since the bacterial strains were already identified, the title should focus solely on the selection of biosurfactant-producing strains.

The title was changed in:

Testing of  agricultural waste as the growth substrate for target biosurfactant-producing microorganisms

6

Results

In Table 6, the title should be more specific, clearly indicating that the P. aeruginosa MAD10 strain was used (if applicable).

The title was changed in:

Formulation of corn-chaff based media for the production of  rhamnolipids by P. aeruginosa MAD10. Values reported are the average of at least 3 independent experiments with a SD below 5%.

7

Results

What is the composition of the industrial medium used? Including this information would be beneficial.

The composition was included in Table 6

8

Results

In section "3.7. Purification of rhamnolipids from medium BCS388 and identification of the different 544 congeners," optimization of biosurfactant production is mentioned. However, I have only observed the selection of carbon sources, and no response surface assays or other experiments identifying optimal parameters are presented. This section should be revised.

We have modified the section according to the suggestion the statement:

“After the optimization of the fermentation conditions for rhamnolipids production” was removed

9

conclusions

The conclusions should focus on the selection of bacteria capable of producing biosurfactants using only agricultural residues as sources of carbon and nitrogen. This becomes complicated when glycerol is added to the fermentation processes. I recommend improving the conclusions based on these points while highlighting the results obtained.

We have modified the discussion according to the suggestion

Reviewer 3 Report

Comments and Suggestions for Authors

The manuscript is well constructed and has clear and detailed writing. However, some points need to be revised to improve the quality of the work.

1-     The English is fine, but there are too many long sentences. Please, consider breaking some of them for better reliability. 

2-     Authors should highlight the focus and relevance of their work.

3-     The abstract should contain some of the findings of the work 

4-     In lines 40-43, it was mentioned that diverse treatments are used to convert agro-wastes into substrates for microbial fermentation. The author should briefly describe some examples of these methods to provide more robust information.

5-     Lines 308-309. The authors report that the determination of residue compositions is described in the materials and method section, but the information is not provided.

6-     In Table 5, the production of biosurfactants in different substrates is indicated by "+" or "-". It is not clear what "+" or "-" means. It should be explained.

7-     Lines 550-552. It is not explained why the method chosen for the extraction of rhamnolipids is considered an environmentally friendly 

8-     Lines 554-555. "The mechanism involved is related to the salting out effect already described in the literature [45]". I recommend describing what is the effect of salting out associated with the extraction of rhamnolipids.

9-     In Subtopic 3.2, I recommend removing the first mention of Table 5. 

10-  Line 437. Replace “see below” for “Table 6”

11-  Line 92. Insert space 

12-  Line 14. Drawbacks 

Comments on the Quality of English Language

The English Language if fine, but needs to be improved. 

Author Response

Dear Reviewer,

we thank you very much for critically revising our manuscript. Below are the answers (in italics) to your questions/comments.

  • The English is fine, but there are too many long sentences. Please, consider breaking some of them for better reliability. 

We have shortened the sentences all along the text. The manuscript was revised by a native English speaker

  • Authors should highlight the focus and relevance of their work.

Focus and relevance of the work was highlighted in the final part of the introduction and in the discussion sections

  • The abstract should contain some of the findings of the work 

We have modified the abstract by adding the main finding of the work which is the production of high amounts of rhamnolipids with a corn-chaff based medium

  • In lines 40-43, it was mentioned that diverse treatments are used to convert agro-wastes into substrates for microbial fermentation. The author should briefly describe some examples of these methods to provide more robust information.

A detailed description of the different pre-treatments was added to the introduction section

5-     Lines 308-309. The authors report that the determination of residue compositions is described in the materials and method section, but the information is not provided.

We are sorry for the mistake. We have changed the sentence:

“The composition of the agro-wastes was determined as described in the section materials and methods and is reported in Table S1.”

With

“The composition of the agro-wastes is reported in Tables S1 to S3 and was determined according to the methods reported in the “Reference or source” column of the same Tables.”

6-     In Table 5, the production of biosurfactants in different substrates is indicated by "+" or "-". It is not clear what "+" or "-" means. It should be explained.

We have added to the Table 5 legend that the "+" or "-" refers to the ODA test positivity or negativity

7-     Lines 550-552. It is not explained why the method chosen for the extraction of rhamnolipids is considered an environmentally friendly 

We have elucidated this concept in the text as follows:

At present, rhamnolipids are extracted from the fermentation broth with methods using organic solvents (such as ethyl acetate, chloroform, dichloromethane) which are risky to handle, aggressive to the environment, and expensive to dispose [26,62]. The goal of this trial was to identify an efficient method to extract rhamnolipids using environmental friendly solvents. The extraction/purification system used, was therefore an alcohol-inorganic salt system.  

8-     Lines 554-555. "The mechanism involved is related to the salting out effect already described in the literature [45]". I recommend describing what is the effect of salting out associated with the extraction of rhamnolipids.

We have added to the text the following description:

“The salting out effect makes the water phase and the 2-propanol phase (which are normally miscible) immiscible. Therefore, the two phases are easily separated, with rhamnolipids localized in the 2-propanol phase.”

9-     In Subtopic 3.2, I recommend removing the first mention of Table 5. 

Mention was removed as suggested

10-  Line 437. Replace “see below” for “Table 6”

We have replaced as suggested

11-  Line 92. Insert space 

 We have inserted as suggested

12-  Line 14. Drawbacks 

We can not see drawback at line 14. We have checked the drawbacks word all along the text and made modifications where pertinent

Comments on the Quality of English Language

The English Language if fine, but needs to be improved. 

We have improved the language, focusing on shortening the sentences all along the manuscript

Round 2

Reviewer 1 Report

Comments and Suggestions for Authors

-